# The Effect of a Coordinative Training in Young Swimmers’ Performance

**DOI:** 10.3390/ijerph19127020

**Published:** 2022-06-08

**Authors:** Ana F. Silva, Pedro Figueiredo, João P. Vilas-Boas, Ricardo J. Fernandes, Ludovic Seifert

**Affiliations:** 1Centre of Research, Education, Innovation and Intervention in Sport, Faculty of Sport, University of Porto, 4200-450 Porto, Portugal; jpvb@fade.up.pt (J.P.V.-B.); ricfer@fade.up.pt (R.J.F.); 2Porto Biomechanics Laboratory, University of Porto, 4200-450 Porto, Portugal; 3Portugal Football School, Portuguese Football Federation, 1495-433 Oeiras, Portugal; pedfig@me.com; 4Faculty of Sport Sciences, University of Rouen Normandy, CETAPS EA3832, 76821 Mont Saint Aignan, France; ludovic.seifert@univ-rouen.fr; 5Institut Universitaire de France (IUF), 75231 Paris, France

**Keywords:** coordination, youth, motor adaptability, ecological dynamics, biomechanics

## Abstract

This study investigated the effects of a coordinative in-water training. Total 26 young swimmers (16 boys) were divided in a training group (that performed two sets of 6 × 25-m front crawl, with manipulated speed and stroke frequency, two/week for eight weeks) and a control group. At the beginning and end of the training period, swimmers performed 50-m front crawl sprints recorded by seven land and six underwater Qualisys cameras. A linear mixed model regression was applied to investigate the training effects adjusted for sex. Differences between sex were registered in terms of speed, stroke length, and stroke index, highlighting that an adjustment for sex should be made in the subsequent analysis. Between moments, differences were noticed in coordinative variables (higher time spent in anti-phase and push, and lower out-of-phase and recovery for training group) and differences between sex were noticed in performance (stroke length and stroke index). Interactions (group * time) were found for the continuous relative phase, speed, stroke length, and stroke index. The sessions exerted a greater (indirect) influence on performance than on coordinative variables, thus, more sessions may be needed for a better understanding of coordinative changes since our swimmers, although not experts, are no longer in the early learning stages.

## 1. Introduction

Performance of locomotor tasks, such as walking, running, or swimming, requires the coordination of spatiotemporal patterns of upper and lower limbs muscle activity [1]. Following the dynamical system approach, coordination will be the result of the interaction among organismic (physical, psychological, morphological, and physiological), task (specific to the task to perform and related to the goal and rules that governing the task), and environment (those that are external to the movement system as light, temperature, or altitude) constraints [2]. Therefore, to achieve a certain goal or performance, a constant management of those constraints must be carried out, as they limit the performers’ action. Therefore, and within the ecological dynamics’ framework, there is no ideal motor coordination solution towards which all learners should aspire, but rather functional coordination patterns that arise from a self-organization [2,3,4,5].

During sport, if more functional movement patterns emerge as a result of movement variability due to the constant interacting constraints, a great movement exploration could be provided, allowing the performer to search for more varied and effective movement solutions to fit task dynamics [6]. It has been argued that presenting the relevant constraints during the different skill development phases is the key for learners acquiring functional movement behaviour [6,7]. In fact, through the analysis of different constraint-led approach in swimming, it was observed that changes in behaviour raised during action e.g., [8,9,10], but fluctuations in the movement patterns themselves may or may not be functional [6], increasing the constant importance of manipulating constraints during the learning process [5].

In swimming, especially in the front crawl technique, different constraints have been analysed to understand how they influence coordination changes. Regarding environmental constraints, studies have been analysing different speeds e.g. [11,12,13,14] and when using some equipment as paddles, swim suits, and parachutes (e.g., [15,16,17]), meaning that swimmers have to face different drag magnitudes. Using task constraints, the influence of breathing action was studied [18,19,20]. However, it is with organismic constraints that more studies have been done, using characteristics or variables related (directly or indirectly) to physical (stroke frequency—SF—and stroke length—SL; e.g., [21,22]), physiological (fatigue and energy cost—e.g., [23,24]) and morphological (sex—e.g., [9,25]) aspects.

In the above-mentioned studies, only one conducted a longitudinal analysis, but with adult swimmers [20], there is an existing lack of studies in young swimmers, who are still in the learning process. In fact, it was found that age changes the physical capacities and, therefore, the athletes’ performance considerably [26], and, after the puberty period, slight coordination increases occur [27]. Nevertheless, it is still difficult to determine the best age for motor learning, but the predispositions seem best up to early adulthood [27]. There is no coordination related interventional studies in young front crawl swimmers and only few exist that characterise young swimmers’ patterns analysing characteristics or variables related (directly or indirectly) to physical (SF and SL) and morphological (anthropometry, maturation, and sex) aspects [10,28,29,30,31].

In a continuous movement conducted only with fingers, it was observed that when movement frequency increases (and, consequently, speed), the coordination mode becomes unstable, only to be replaced by another stable mode (e.g., [32]). In fact, this influence was also observed in adult studies when swimming front crawl, suggesting that speed and SF were the major influencing factors on changes in front crawl swimming coordination. The aim of this study was to develop an eight-weeks coordinative training in young swimmers to investigate if more stable modes emerged at the fastest front crawl race (50-m). It was hypothesized that swimmers included in the coordinative training expressed a great coupled between upper limbs after the intervention, comparing to control group.

## 2. Materials and Methods

### 2.1. Participants

Total 26 young swimmers (16 boys), free from injury and training, at least, six times/week, participated in the current study. Participants were divided into two groups. Considering that they belonged from two different teams, one team was considered the control (CG) and the other team, the in-water coordinative training group (TG). Nevertheless, the entire sample compete at the same level and their training characteristics were similar. To be included in this study, swimmers should have participated in at least 70% of complementary training sessions (11 sessions) and in the two measuring moments. Following this, three (all boys) and four swimmers (three boys and one girl) were excluded from the CG and TG, respectively. Table 1 shows the main physical and training background characteristics of each group separated by sex. The local ethics committee approved the procedures and all the swimmers’ parents signed a consent form in which the protocol was explained. In addition, a maturation evaluation was accomplished, concluding that all swimmers were in the post-pubertal maturational stage (stage 4 or higher; [33]).

### 2.2. Training Sessions

To obtain a larger group of swimmers, the different training sessions were performed with two different teams, but their coaches followed a similar swimming training plan (regarding frequency, volume, and intensity). The CG only performed the normal training sessions in the swimming pool (without any complementary sessions) and the TG performed an additional in-water coordinative training session two/week for eight weeks (total 16 sessions). Those coordinative training were supervised by the first the first author.

### 2.3. In-Water Coordinative Training Sessions

Considering that speed and SF were the main influencing factors in front crawl swimming technique [12,22], it was used a task manipulation focusing on these two variables. However, as the aim of the current study was also to understand the impact of coordination on performance, the maximal speed was also the target of the in-water coordinative training. Therefore, two sets of 6 × 25-m maximal speed front crawl [34] were implemented in each training session and SF was manipulated in each repetition as follow: (i) preferred SF; (ii) slightly lower SF; (iii) greatly lower SF comparing to the preferred one; (iv) preferred SF; (v) slightly higher SF; and (vi) greatly higher SF comparing to the preferred one. In each repetition the individual SF and the respective speed value was registered and feedback was given to swimmers. The maximal speed was measured in the pre-test as well as their preferred SF, using a stopwatch. During the training sessions, the time at 25-m was always measured (individually) and each swimmer was asked the number of upper limb cycles performed. The interval between each repetition was established at 1 min and between sets was 3 min.

### 2.4. Test Procedures

One month before starting the intervention program, swimmers performed the same pre and post-tests evaluations, trying to understand if the training characteristics were similar between groups (since they belong to two different teams). A standardized 1000 m warm-up at low to moderate swimming intensity was conducted in a 25-m indoor pool before experiments. Afterwards, each swimmer performed a self-paced 50-m front crawl at maximal speed, starting in-water (without diving), with a non-breathing pattern in the centre of the pool to avoid start, turn, and breathe effects on coordination. After each trial, participants were informed of their performance and if their time was not within ±2.5% of their 50-m race time, he/she repeated the trial (to ensure that swimmers performed at their best).

### 2.5. Apparatus

While performing the 50-m front crawl test, swimmers used 10 anatomical reflective landmarks in each body side (iliac crest, acromion, lateral humerus epicondyle, and radius- and ulnar-styloid processes), enabling a 3D dual media working volume creation, where the orthogonal axes were defined as x, y, and z for horizontal, medio-lateral, and vertical (z = 0 defines the water surface) movements, respectively. A 13 camera setup (MoCap) was used, with seven land plus six underwater cameras (Oqus 3+ and Oqus Underwater, Qualisys AB, Gothenburg, Sweden) operating at 100 Hz. The calibrated volume was defined using underwater, above water, and twin system to merge the first and the latter calibrations (according to the manufacturer’s guidelines).

### 2.6. Biomechanical Variables

Swimming speed was assessed through the ratio of the hip displacement in an upper limbs cycle (distance travelled between two consecutive entries of the same hand) to its total duration. SL was determined by the horizontal distance travelled by the hip during an upper limbs cycle and SF was determined as the number of cycles performed per minute. Stroke index (SI) was computed by the product of speed and SL, and intra-cyclic velocity variation (IVV) was calculated through the ratio between speed standard deviation to mean speed.

### 2.7. Upper-Limbs Coordination Analysis

Coordination between right and left upper limbs was assessed through the continuous relative phase (CRP) [35,36]. CRP assessment between upper limbs (arm–shoulder–trunk angle) was performed for two upper limbs cycles, recorded in the central part of the pool, with cycle duration expressed in percentage allowing its comparison. The CRP was calculated through the subtraction of the phase angle of the two oscillators at each point in time over the entire cycle (i.e., the left shoulder phase angles were subtracted from the right one). CRP values can range from 0–360°, but a variation of ± 30° was accepted for the determination of a coordination pattern [37,38,39]. Therefore, three different modes could be found: in-phase (when 330° < CRP < 30°), anti-phase (when 150° < CRP < 210°) and out-of-phase (when 30° < CRP < 150° and 210° < CRP < 330°). From that analysis, different variables were extracted to examine the coordination between upper limbs: (i) the mean CRP and its variability through the Standard Deviation of CRP (SD of CRP) over a cycle; and (ii) the relative time spent in in-phase, out-of-phase, and in anti-phase (all expressed in %) to inform about the coupling between upper limbs coordination.

The relative time between two propulsive upper limbs actions was also calculated, namely the index of coordination (IdC; [12]), characterized as the time between the beginning of propulsion of the first right and the end of propulsion of the first left upper limbs actions, and between the beginning of propulsion of the second left and the end of propulsion of the first right upper-limbs actions. IdC was calculated based on the division of the upper limbs actions in four phases: (i) entry and catch, corresponding to the time since the entry of the hand in-water until it starts to make the backward movement; (ii) pull, since the end of the previous action until achieve the vertical alignment of the shoulder (first propulsive phase); (iii) push, since the end of the previous action to the exit the hand of the water (second propulsive phase); and (iv) recovery, covering the time from the exit of the hand until its new entry. The IdC and each cycle phase were expressed as the percentage of the duration of a complete upper limbs cycle and the sum of pull and push phases, and of catch and recovery phases, indicate the duration of propulsive and non-propulsive phases, respectively [12]. Three different synchronisation modes are possible to identify in front crawl: (i) opposition (IdC = 0%), when one upper limb begins the propulsive phase and the other is finishing it, providing continuous motor action; (ii) catch-up (IdC < 0%), existing a lag time between propulsive phases of the two upper limbs; and (iii) superposition (IdC > 0%), describing an overlap in the propulsive phases of both upper limbs.

### 2.8. Statistical Analysis

To understand if experimental groups had similar training characteristics, an analysis was conducted for the pre-test and the tests performed one month before. All the statistical analysis were conducted with linear mixed models adjusted for sex (a widely used method for longitudinal continuous data that considers correlation between repeated measures and the maximum likelihood estimators are easily obtained using standard software [40]). Changes in groups over time (group * time interaction) in coordinative (CRP, standard deviation of CRP, in-phase, anti-phase, out-of-phase, IdC, the four upper limbs phases, propulsive, and non-propulsive phases) and performance variables (speed, SF, SL, SI, and IVV) were modelled using a linear mixed model regression with random-effects statements on intercept of each participant. Adjustments for sex were conducted in all variables analysed. The covariance type used for the random effects was the unstructured option (completely general covariance matrix). Normality of residuals was visually verified and data were expressed as mean ± SD. Values of *p* < 0.05 were considered significant and tests were two-sided, with statistical analysis performed using IBM SPSS software version 24.0 (SPSS, Chicago, IL, USA).

## 3. Results

Comparison between pre-test and the tests performed one month before, did not report significant differences in any variable included in the current study, neither between groups, nor time or their interaction, thus the differences registered in the post-test could be related to the implemented trainings. Nevertheless, that analysis provided some information about the sex effect, since in some performance variables (speed, SL, and SI) a significant effect was found.

### 3.1. The Effect of Training on Coordinative Variables

The implemented coordinative training showed no influence on standard deviation of CRP, time percentage spent in in-phase, IdC, time percentage spent in entry and catch, propulsive, and non-propulsive phases. However, as presented in Table 2 and Table 3, in the time spent in anti-phase, out-of-phase, and push phase differences were noticed from the pre- to the post-test (models 4, 5, and 8, respectively). Furthermore, a significant group*time interaction for CRP (Table 2 and Figure 1) was observed.

### 3.2. The Effect of Training on Performance Variables

As shown in Table 4 and Figure 2, time exerted a relevant effect in all performance variables (except speed) and a sex effect on speed, SL, and SI was observed. Moreover, significant group * time interactions for speed, SL, and SI (Table 4 and Figure 2) were detected.

## 4. Discussion

### 4.1. The Effect of Training on Coordinative Variables

In the current study, two different methods were used to analyse front crawl swimming coordination, the CRP and its components (standard deviation of CRP, in-, anti-, and out-of-phase) and IdC and its upper limbs phases, with the first exhibiting information about space and time, and the later only providing time-related information. Nevertheless, few differences between moments were noticed in both methods, with distinct behavioural trends. The TG slightly increased the time spent in anti-phase, while presented slight declines in out-of-phase, in opposition to the CG, suggesting that the first swimmers tried to stabilize their swimming pattern, since these phases changed less comparing to the CG. Although not different, the standard deviation of CRP seemed to confirm that the TG is stabilizing a pattern, showing a trend to decrease its value in opposition to the CG swimmers. Nevertheless, at the end of the intervention, CRP expressed a mean value related to an anti-phase mode in both groups (as the swimming technique demands), but the interaction found (group * time) denote that those groups developed different coordinative profiles along the time because of the coordinative trainings conducted. Although not too expressive, those results are in line with the hypothesis made, which expected a greater coupling between upper limbs after the intervention in the TG.

It is known that in the fastest races, younger swimmers are used to present lower time percentage spent in pull phase (~18%; e.g., [28,31]) comparing to adults (~24%; e.g., [12]), but it would not be expected decreases in that upper limbs phase during the current intervention. In fact, both groups decreased that phase, with the TG showing a greater drop. Considering that speed trend to rise in TG, these values seem to suggest that swimmers that participated in the intervention program applied better their propulsive phases, although they showed lower results in the propulsive phases alone and, consequently, in its sum. Indeed, the lower IVV registered by TG in the post-test seems to confirm that idea, since a low IVV has been related to higher propulsive continuity (e.g., [41]) and higher swimming efficiency [42]. Hence, as suggested above, these could be the result of an inappropriate hand orientation by the CG swimmers [43,44], leading them to spend more time in that upper limbs phase.

Following the ecological approach, a constraint-led perspective allows to search for functional coordination solutions that arise from the individual [2,45], leading to discover preferred and typically stable coordination patterns [46,47]. Empirical evidence in sport showed that when informational task constraints is altered, different patterns tend to arise [48], but it could occur a continued pattern improvement instead of an emergence of a new one [46]. This could had happened in the current study, since our swimmers have already more than five years of practice, and they could be between control or skill stage [2] where a stable coordinative front crawl pattern already exist. Notwithstanding, the training program implemented was not enough to observe IdC changes toward the coordinative patterns adopted by adult swimmers that in the fastest races achieved IdC values closer or above zero (superposition only in men; [12,22]).

Studies with young swimmers [10,31], found that with similar coordination framework, the coordination differences among groups were found in the contents of relevant components and kinematic variables, corroborating our findings. Based on non-linear dynamics, studies showed that when analysing coordination changes, it must be considered the perturbation magnitude of the existing constraints, distinguishing low- and high-order behaviour variables. The former is usually related to general biomechanical variables (e.g., speed and SF), reflecting simple inherent mechanisms (i.e., over space or time) that lead to the emergence of behaviour and the later combine multiple lower-order variables to capture the system coordination dynamics [49]. Therefore, the locomotor system seems to use a rich repertoire of compensatory adjustments in response to different task and environmental constraints [49,50].

Furthermore, although no significant decreases in the standard deviation of CRP were observed, a trend to show different behaviour is displayed in Figure 1, with the TG exhibiting a trend to decrease and the CG to increase. In fact, the sample age could have influenced those results, since it is known that the movement variability follows the central nervous system development [51,52], showing a decrease through childhood and adolescence to adulthood [53]. This fact seems to explain why the standard deviation of CRP and IdC results did not altered significantly, suggesting that the coordinative pattern adopted by young swimmers also depend on their central nervous system maturation. Still, the currently applied training program seemed to enlarge the TG coordinative repertoire, as swimmers increased (slightly) speed and SL and the propulsive upper limbs phases seemed to be more efficient. Indeed, it was suggested that the lower range of swimming speeds during training and race could result in a lower range of coordination repertoire, for instance, by long-distance swimmers and triathletes [54,55].

### 4.2. The Effect of Training on Performance Variables

Differences between pre- and post-tests were noticed in SL, suggesting that both groups showed technical improvements. In fact, it is known that an improved swimming technique results in a longer SL, with this variable being more related to performance than SF [54,56,57]. Moreover, a SF minimization while increasing SL from the first to the second evaluation moment was noticed, which is a strategy used by elite swimmers to attain a more economical swimming pattern [56,58]. Considering that speed results in the product between SL and SF [59], the greater SL increase in TG comparing to CG explains the trend to exhibit a superior speed by TG at the post-test.

Following the above-referred changes, and confirming that a technical improvement occurred, SI (a swimming efficiency related variable [60,61,62]) registered better results in the post-test. However, similarly to SL and speed values, this SI improvement (once it is the product between speed and SL) was mainly due to changes occurred in the TG, which obtained higher results in the post-test. In addition, IVV decreased from pre- to post-test, even if in opposition to the SI trend, with similar evolution patterns being observed since both groups decreased it. This last variable has been also considered an indicator of swimming efficiency (e.g., [23,63]), since swimming speed varies as a result of accelerations and decelerations due to the propulsive forces application [64,65]. In fact, it was found that a 10% speed chances within an upper limbs cycle results in an ~3% additional work demand, highlighting that lower IVV values could be the best key to increase the capacity to produce propulsive forces while minimizing power output [62].

The eight-weeks training program coincided mainly with the specific preparation phase that is generally characterized as a period of a volume and intensity training increases [66], leading to possible maximal speed reductions [67]. In fact, a slight speed decline was noticed in CG, in opposition to the TG, which could be due to the fact that the training program chosen mainly focused at speed development [34]. Therefore, the TG had two extra speed training sessions/week, which could have been enough to slightly increase this motor capacity. The interactions found in SL and SI explain the speed described results, evidencing that TG swimmers (boys and girls) experienced greater technical improvements due to the specific coordinative training program. Indeed, upper limbs coordination is not only important for perfecting propulsive phases, but also to cover the buoyancy and breathing issues [43].

This study had some limitations. First, we could argue that those swimmers were not from the same team and same training group, however, those teams were chosen because they were both from the same level, they belonged to clubs that compete at national level, and their coaches follow a similar training plan. In addition, an initial comparison was made between the two groups, considering the variables included in this study, and differences were only found between sexes, but not between groups. Secondly, the fact that the TG slightly increase their training volume with the coordinative sessions could led to link the improvement to this extra volume of training, nevertheless, that extra was really too small, since those swimmers were used to a volume of around 16 h of training per week, thus an extra of around 30 min (corresponding to the two extra coordinative trainings) did not seem to be significant.

## 5. Conclusions

Considering that speed and SF were the previously described most relevant influencing factors of front crawl swimming coordination [12,22], speed could also be manipulated (not only SF). However, since we were analysing competitive swimmers and the study target was also performance related, we have always used maximal swimming speed, excluding environmental oscillations (resulting from drag variations). In addition, there is a possibility that the reduced number of young swimmers composing our sample could have influenced the current study results. Nevertheless, conducting experimental procedures in an aquatic environment is a hard task to be accomplished and, considering that our sample was composed of competitive swimmers representing two well-organized swimming clubs, we can assume that it is representative of young swimmers’ reality.

## Figures and Tables

**Figure 1 ijerph-19-07020-f001:**
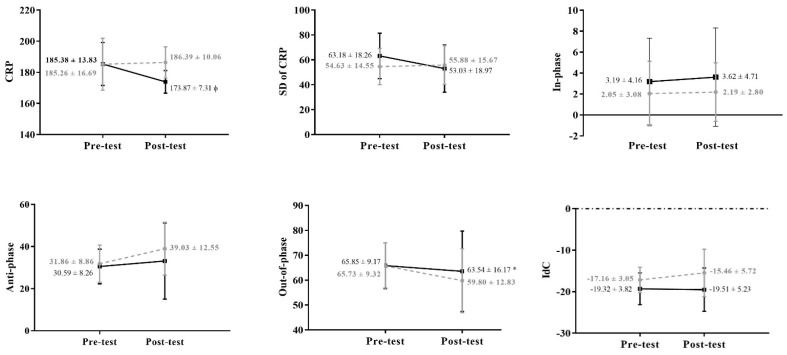
Pre and post-test comparisons for coordinative variables: continuous relative phase (CRP), standard deviation of the continuous relative phase (SD of CRP), in-phase, anti-phase, out-of-phase, index of coordination (IdC), and the four upper-limb phases (entry and catch, pull, push, and recovery phases), with the grey and black lines representing the control and the training groups, respectively. The symbols * mean that a time effect was observed and φ that an interaction time x group effect was found.

**Figure 2 ijerph-19-07020-f002:**
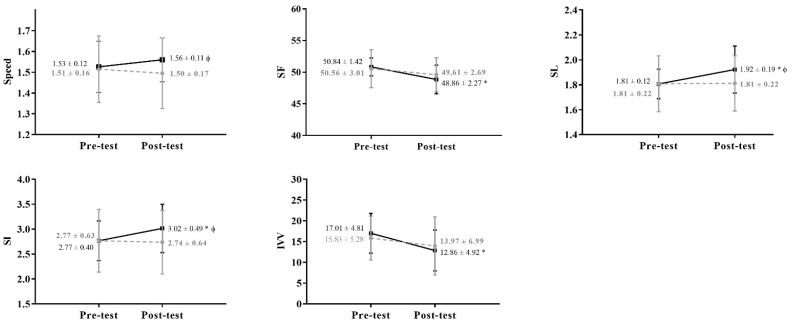
Pre and post-test comparisons for performance variables: speed, stroke frequency (SF), stroke length (SL), stroke index (SI), and intra-cyclic velocity variations (IVV), with the grey and black lines representing the control and the training groups, respectively. The symbols * mean that a time effect was observed and φ that an interaction time × group effect was found.

**Table 1 ijerph-19-07020-t001:** Age, height, body mass, and training background characteristics of the control and coordinative training groups.

	Control Group (n = 11)	Training Group (n = 8)
Age (years)	14.8 ± 0.9	14.8 ± 0.7
Height (cm)	166.6 ± 0.1	170.3 ± 0.1
Arm span (cm)	167.1 ± 0.2	173.0 ± 0.1
Body mass (kg)	58.3 ± 10.1	58.1 ± 10.2
Swimming practice (years)	5.5 ± 1.0	5.5 ± 0.9

**Table 2 ijerph-19-07020-t002:** Linear mixed model regression for continuous relative phase (CRP), standard deviation of continuous relative phase (SD of CRP), in-phase, anti-phase, out-of-phase, and index of coordination (IdC), with the unadjusted model and adjusted for sex (models 1–6, respectively).

			Slope (SE); Statistical Inference
CRP	Unadjusted model	Group	12.78 (9.95); *p* = 0.21
		Time	1.14 (2.99); *p* = 0.71
		Group * Time	−12.65 (4.61); *p* = 0.01
	Model 1	Group	13.33 (9.84); *p* = 0.19
		Time	0.54 (3.89); *p* = 0.89
		Group * Time	−12.74 (4.60); *p* = 0.01
		Sex	−6.94 (9.84); *p* = 0.49
SD of CRP	Unadjusted model	Group	19.96 (12.02); *p* = 0.11
		Time	1.26 (4.58); *p* = 0.79
		Group * Time	−11.41 (7.06); *p* = 0.12
	Model 2	Group	20.23 (12.03); *p* = 0.11
		Time	0.47 (5.99); *p* = 0.94
		Group * Time	−11.52 (7.08); *p* = 0.12
		Sex	−3.44 (12.03); *p* = 0.78
In-phase	Unadjusted model	Group	0.88 (2.57); *p* = 0.28
		Time	0.17 (0.96); *p* = 0.86
		Group * Time	0.26 (1.48); *p* = 0.86
	Model 3	Group	0.73 (2.54); *p* = 0.78
		Time	1.14 (1.20); *p* = 0.36
		Group * Time	0.40 (1.42); *p* = 0.78
		Sex	1.90 (2.54); *p* = 0.46
Anti-phase	Unadjusted model	Group	3.31 (8.00); *p* = 0.82
		Time	7.17 (4.28); *p* =0.34
		Group * Time	−4.57 (6.60); *p* = 0.61
	Model 4	Group	2.44 (7.67); *p* = 0.77
		Time	12.89 (5.22); *p* = 0.03
		Group * Time	−3.74 (6.16); *p* = 0.56
		Sex	10.87 (7.67); *p* = 0.23
Out-of-phase	Unadjusted model	Group	−3.51 (8.91); *p* = 0.70
		Time	−5.93 (4.43); *p* = 0.19
		Group * Time	3.62 (6.83); *p* = 0.60
	Model 5	Group	−2.44 (8.05); *p* = 0.76
		Time	−12.99 (4.72); *p* = 0.01
		Group * Time	2.60 (5.58); *p* = 0.65
		Sex	−13.45 (8.05); *p* = 0.11
IdC	Unadjusted model	Group	−0.26 (2.08); *p* = 0.90
		Time	1.70 (1.10); *p* = 0.14
		Group * Time	−1.89 (−1.11); *p* = 0.28
	Model 6	Group	−0.23 (2.09); *p* = 0.92
		Time	2.17 (1.43); *p* = 0.15
		Group * Time	−1.83 (1.69); *p* = 0.30
		Sex	−0.45 (2.09); *p* = 0.83

**Table 3 ijerph-19-07020-t003:** Linear mixed model regression for entry and catch, pull, push, recovery, propulsive, and non-propulsive phases, with the unadjusted model and adjusted for sex (models 7–12, respectively).

			Slope (SE); Statistical Inference
Entry and catch	Unadjusted model	Group	2.89 (4.48); *p* = 0.54
		Time	2.53 (2.26); *p* = 0.28
		Group * Time	0.79 (3.48); *p* = 0.83
	Model 7	Group	2.44 (4.31); *p* = 0.60
		Time	4.33 (2.89); *p* = 0.16
		Group * Time	1.05 (3.41); *p* = 0.76
		Sex	5.62 (4.31); *p* = 0.25
Pull	Unadjusted model	Group	0.84 (2.77); *p* = 0.77
		Time	−2.21 (1.38); *p* = 0.13
		Group * Time	−1.58 (2.13); *p* = 0.47
	Model 8	Group	1.04 (2.72); *p* = 0.71
		Time	−3.13 (1.78); *p* = 0.10
		Group * Time	−1.72 (2.10); *p* = 0.43
		Sex	−2.50 (2.72); *p* = 0.39
Push	Unadjusted model	Group	0.78 (2.40); *p* = 0.76
		Time	4.87 (1.32); *p* = 0.00
		Group * Time	−2.17 (2.04); *p* = 0.31
	Model 9	Group	0.93 (2.37); *p* = 0.72
		Time	4.45 (1.72); *p* = 0.03
		Group * Time	−2.24 (2.04); *p* = 0.30
		Sex	−1.82 (2.37); *p* = 0.49
Recovery	Unadjusted model	Group	−5.70 (3.98); *p* = 0.18
		Time	−5.64 (1.81); *p* = 0.01
		Group * Time	3.42 (2.78); *p* = 0.23
	Model 10	Group	−5.61 (3.99); *p* = 0.18
		Time	−5.95 (2.36); *p* = 0.02
		Group * Time	3.38 (2.79); *p* = 0.24
		Sex	−1.10 (3.99); *p* = 0.79
Propulsive Phases	Unadjusted model	Group	1.63 (3.16); *p* = 0.64
		Time	2.66 (1.78); *p* = 0.17
		Group * Time	−3.76 (2.75); *p* = 0.20
	Model 11	Group	1.97 (2.75); *p* = 0.48
		Time	1.31 (1.72); *p* = 0.45
		Group * Time	−3.95 (2.03); *p* = 0.06
		Sex	−4.33 (2.75); *p* = 0.13
Non-propulsive phases	Unadjusted model	Group	−1.36 (3.18); *p* = 0.69
		Time	−2.38 (1.75); *p* = 0.20
		Group * Time	3.48 (2.70); *p* = 0.22
	Model 12	Group	−1.73 (3.01); *p* = 0.61
		Time	−0.85 (2.23); *p* = 0.71
		Group * Time	3.71 (2.63); *p* = 0.19
		Sex	4.68 (3.01); *p* = 0.22

**Table 4 ijerph-19-07020-t004:** Linear mixed model regression for speed, stroke frequency (SF), stroke length (SL), stroke index (SI), and intra-cyclic velocity variations (IVV), with the unadjusted model and adjusted for sex (models 13–17, respectively).

			Slope (SE); Statistical Inference
Speed	Unadjusted model	Group	−0.04 (0.07); *p* = 0.58
		Time	−0.02 (0.01); *p* = 0.04
		Group * Time	0.05 (0.01); *p* = 0.00
	Model 13	Group	−0.02 (0.03); *p* = 0.62
		Time	−0.02 (0.01); *p* = 0.05
		Group * Time	0.05 (0.01); *p* = 0.00
		Sex	−0.25 (0.03); *p* = 0.00
SF	Unadjusted model	Group	1.30 (1.71); *p* = 0.46
		Time	−0.95 (0.61); *p* = 0.14
		Group * Time	−1.03 (0.95); *p* = 0.29
	Model 14	Group	1.44 (1.67); *p* = 0.40
		Time	−1.48 (0.78); *p* = 0.07
		Group * Time	−1.11 (0.92); *p* = 0.25
		Sex	−1.70 (1.67); *p* = 0.32
SL	Unadjusted model	Group	−0.12 (0.08); *p* = 0.18
		Time	0.00 (0.02); *p* = 0.89
		Group * Time	0.11 (0.03); *p* = 0.00
	Model 15	Group	−0.10 (0.07); *p* = 0.16
		Time	0.02 (0.02); *p* = 0.36
		Group * Time	0.12 (0.03); *p* = 0.00
		Sex	−0.22 (0.07); *p* = 0.00
SI	Unadjusted model	Group	−0.28 (0.24); *p* = 0.26
		Time	−0.03 (0.03); *p* = 0.38
		Group * Time	0.28 (0.05); *p* = 0.00
	Model 16	Group	−0.21 (0.15); *p* = 0.17
		Time	−0.00 (0.04); *p* = 0.95
		Group * Time	0.28 (0.05); *p* = 0.00
		Sex	−0.80 (0.15); *p* = 0.00
IVV	Unadjusted model	Group	3.45 (3.13); *p* =0.28
		Time	−1.87 (1.23); *p* = 0.15
		Group * Time	−2.28 (1.90); *p* = 0.25
	Model 17	Group	3.41 (3.13); *p* = 0.29
		Time	−2.38 (1.60); *p* = 0.15
		Group * Time	−2.35 (1.89); *p* = 0.23
		Sex	0.57 (3.13); *p* = 0.86

## Data Availability

The data presented in this study are available on request from the corresponding author.

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
