# Peer review of "The Effect of a Coordinative Training in Young Swimmers’ Performance"

_ijerph, 2022, doi:10.3390/ijerph19127020_

Round 1
Reviewer 1 Report
see attachment

Author Response
Attention, doubt: Training Sessions: the cg only performed the normal training 98 sessions in the swimming pool (without any complementary sessions). the tg performed 99 twice a week for 8 weeks (16 sessions) an additional coordinative training session in water. - The control group (CG) performed a smaller training volume, as it did not perform additional coordination exercises in relation to the training group (TG). It should be considered, and maybe checked in subsequent studies, whether the lower load of the control influenza conversion training at this time is not a factor differentiating the results.
Authors: Dear Reviewer, the authors understand your concerns regarding the additional training sessions conducted by the TG. However, the added set was only two sets of 6x25m of front crawl swimming at maximal velocity, which added not even 15 minutes in each swimming session for the TG. Considering that those swimmers normally train at least 100 minutes per session, 8 times per week, the Authors believe that the volume increased in the TG each week was not significant to have given them a high learning advantage. Nevertheless, this issue was included in the limitations session.
In the study, which proves its high substantive value, the following are described in detail: In-water coordinative training sessions, Test procedures, Apparatus. Biomechanical parameters, Upper-limb coordination analysis
These procedures are correct and prove a great knowledge of research issues. The research was carried out by recording the tested factors by seven land cameras and six Qualisys underwater cameras.
The results are given: Figure 1. Pre and post-test comparisons for speed, stroke frequency (SF), stroke length (SL), stroke index (SI), intra-cyclic velocity variations (IVV), index of coordination (IdC), continuous relative phase (CRP) and standard deviation of the continuous relative phase (SD of CRP), with the grey and black lines representing the control and the training groups, respectively - however, it has not been included in the text and should be completed.
Authors: Thank you for your words. We tried to describe all the procedures included in the evaluations. We apologize for the missing figure. It was added in the present version.
The other results are well presented.
Authors: Thank you for your words.
The whole work is valuable and worth publishing, especially since the authors showed a great knowledge of the subject, supported by 62 bibliographies, therefore I encourage the authors to continue their research and publication on the determinants of speed in swimming. Good job.
Authors: Thank you for your critical analysis and for your words. The Authors will try to improve the knowledge about the coordination mechanisms, specially in the aquatic environment.
Reviewer 2 Report
First of all, congratulations to the authors for the submitted manuscript.
The study is well conducted, addresses all the recommended sections, and provides some proposals for application to training. Some recommendations are suggested.
The aim of the study was to develop an 8-week coordinative training in young swimmers to investigate if more stable modes emerged at the fastest front crawl race (50-m).
1. It is recommended in the abstract section to include and explain in greater detail a) The type of swimming modality analyzed (crawl); b) Include the two groups considered (TG and Control Group); c) The number of coordination sessions implemented by TG and reference to the type of training carried out by the CG; d) Include some final conclusion of the study, etc.
In general, it is recommended to write a more precise and explanatory abstract of the study carried out.
2. In the introductory section, reference is made to space-time coordination, and examples of sports whose management of space-time parameters are not related to swimming are presented. The space-time "fit" and coordinative demands are very different in sports such as sailing, boxing, basketball, and swimming; please justify the relationship, or modify the examples, since the coordination requirements are very different in "open sports, or external regulation" and "closed sports, or internal regulation, such as swimming". The same in the discussion section, lines 271 and following.
3. In the manuscript a hypothesis is included, but in the conclusions there is no reference to whether it is fulfilled or not. Please explain this.
4. It is recommended to explain how the participants were selected in each of the two groups.
5. Line 191, "Values" of P
6. Consider the possibility of presenting figures 2 and 3 in another format that allows a better view and interpretation of the results.
7. The conclusions section is written mostly as a limitations section. It is recommended to write a specific paragraph of limitations.
8. Consider writing the conclusions of the study in the conclusions section, as well as the proposals for practical application, and the result of the hypothesis raised. It seems that this section should be rewritten to provide the "summary" of the findings that are provided by this study.
9. Special attention to the bibliographic references, since very few current ones are provided. The most recent are one from 2016 and two from 2015.
Author Response
First of all, congratulations to the authors for the submitted manuscript. The study is well conducted, addresses all the recommended sections, and provides some proposals for application to training. Some recommendations are suggested.
Authors: Dear Reviewer, thank you for your words and thank you for helping us to improve our work.
The aim of the study was to develop an 8-week coordinative training in young swimmers to investigate if more stable modes emerged at the fastest front crawl race (50-m).
1. It is recommended in the abstract section to include and explain in greater detail a) The type of swimming modality analyzed (crawl); b) Include the two groups considered (TG and Control Group); c) The number of coordination sessions implemented by TG and reference to the type of training carried out by the CG; d) Include some final conclusion of the study, etc.
In general, it is recommended to write a more precise and explanatory abstract of the study carried out.
Authors: Dear Reviewer, thank you for your valuable comments. The Authors made an effort to improve the abstract for a better clarification.
2. In the introductory section, reference is made to space-time coordination, and examples of sports whose management of space-time parameters are not related to swimming are presented. The space-time "fit" and coordinative demands are very different in sports such as sailing, boxing, basketball, and swimming; please justify the relationship, or modify the examples, since the coordination requirements are very different in "open sports, or external regulation" and "closed sports, or internal regulation, such as swimming". The same in the discussion section, lines 271 and following.
Authors: Thank you for your comment. In fact, coordination is different in those sports, as the environment and the position of the body (horizontal in swimming) is different. Therefore, the Authors included studies in swimming to give some examples.
3. In the manuscript a hypothesis is included, but in the conclusions there is no reference to whether it is fulfilled or not. Please explain this.
Authors: Thank you for noting that. It was added in the discussion session.
4. It is recommended to explain how the participants were selected in each of the two groups.
Authors: Dear Reviewer, the two groups were made according to their habitual group of training, i.e. in the whole sample of young swimmers were composed by two teams from two different clubs. However, those team were chosen because they are at the same level and their coaches followed a similar swimming training plan (regarding frequency, volume and intensity.
5. Line 191, "Values" of P
Authors: Thank you for noticing that. It was changed accordingly.
6. Consider the possibility of presenting figures 2 and 3 in another format that allows a better view and interpretation of the results.
Authors: The figures were changed for a better interpretation of the results.
7. The conclusions section is written mostly as a limitations section. It is recommended to write a specific paragraph of limitations.
Authors: Dear Reviewer, a limitation session was added in the discussion session, as suggested.
8. Consider writing the conclusions of the study in the conclusions section, as well as the proposals for practical application, and the result of the hypothesis raised. It seems that this section should be rewritten to provide the "summary" of the findings that are provided by this study.
Authors: Dear Reviewer, thank you for your comment that helped us to improve our manuscript. Changes were made accordingly.
9. Special attention to the bibliographic references, since very few current ones are provided. The most recent are one from 2016 and two from 2015.
Authors: Thank you for noting that. The Authors included more recent literature.
Reviewer 3 Report
In the present study entitled "The effect of a coordinative training in young swimmers’ performance" an attempt was made to influence certain technical characteristics and performance of swimming through a training method.
Comments
99. Who supervised the coordination training sessions and how?
107. Was the swim speed consistent in each repetition? How is slightly or greatly SF defined? Were the extra or fewer cycles per minute or the percentage on the preferred Sf the same for all swimmers?
122. "± 2.5% of their 50-m race time, he/she repeated the trial." What is the issue if swimmers had better results regarding their records? Does the possibility of different results if the initial trial was analyzed exist?
137. SL... it appears for the first time, please write the phrase as Stroke Length (SL)...
139. SI and IVV appear for the first time, please write Stroke Index (SI)... and Intra-cyclic Velocity Variations (IVV)...
143. Change the phrase to... through the continuous relative phase (CRP) [31,32]. CRP assessment between...
154. Please, rewrite the phrase... through the Standard Deviation of CRP (SD of CRP) over a cycle; (ii)...
207. The Tables and Figures are not match
214. Figure 1 is not shown in the PDF
Conclusion
These are thoughts or limitations. A better conclusion and a proposal for future research is needed to be written
Author Response
In the present study entitled "The effect of a coordinative training in young swimmers’ performance" an attempt was made to influence certain technical characteristics and performance of swimming through a training method.
Comments
99. Who supervised the coordination training sessions and how?
Authors: The coordinative training sessions were implemented and supervised by the first author, which has a swimmers’ coach license and accompanied the two teams during the entire season. That information was added in the manuscript.
107. Was the swim speed consistent in each repetition? How is slightly or greatly SF defined? Were the extra or fewer cycles per minute or the percentage on the preferred Sf the same for all swimmers?
Authors: Dear Reviewer, thank you for your questions that help us to better clarify the methods session. In fact, in the pre-tests conducted, the maximal velocity and its respective time was measured for each swimmer. Therefore, when this 8-week training sessions started, each swimmers knew their maximal speed and their SF (preferred and the other four different SF). However, at the beginning of each training session, swimmers were reminded of their target time and SF. At each repetition of the series, the time was recorded, and the number of upper limb cycles performed was asked. Feedback was given to each repetition, individually, according to the target and achieved speed and SF. Considering that these swimmers are used to executing this type of training strategy (speed and upper limb as a target), the asked task was easily performed by the swimmers.
122. "± 2.5% of their 50-m race time, he/she repeated the trial." What is the issue if swimmers had better results regarding their records? Does the possibility of different results if the initial trial was analyzed exist?
Authors: Dear Reviewer, in fact those swimmers train every day to improve their best time, but if they improve or decrease from one repetition from another, it could indicate that they did not perform their best when asked. Therefore, this interval guarantees that the swimmers did their best. However, this value was referenced at each moment of evaluation, i.e., it does not mean that the value was watertight throughout the season.
137. SL... it appears for the first time, please write the phrase as Stroke Length (SL)...
Authors: Thank you for noticing that. The authors added in the introduction session when this variable appears from the first time.
139. SI and IVV appear for the first time, please write Stroke Index (SI)... and Intra-cyclic Velocity Variations (IVV)...
Authors: Thank you for noticing that. It was added as suggested.
143. Change the phrase to... through the continuous relative phase (CRP) [31,32]. CRP assessment between...
Authors: It was changed as suggested.
154. Please, rewrite the phrase... through the Standard Deviation of CRP (SD of CRP) over a cycle; (ii)...
Authors: It was changed as suggested.
207. The Tables and Figures are not match
Authors: Thank you for noting that. It was revised as suggested.
214. Figure 1 is not shown in the PDF
Authors: The authors apologize for that lapse. In this new version, the figures have been changed as suggested by the second reviewer.
Conclusion
These are thoughts or limitations. A better conclusion and a proposal for future research is needed to be written
Authors: Thank you for noticing that. A limitation session was added at the end of the discussion session and the conclusion session was rewritten as suggested.